# On the Representational Efficiency of Restricted Boltzmann Machines

**James Martens**[*]   **Arkadev Chattopadhyay**[+]   **Toniann Pitassi**[*]   **Richard Zemel**[*]

[*]Department of Computer Science
University of Toronto
{jmartens,toni,zemel}@cs.toronto.edu

[+]School of Technology & Computer Science
Tata Institute of Fundamental Research
arkadev.c@tifr.res.in

## Abstract

This paper examines the question: What kinds of distributions can be efficiently represented by Restricted Boltzmann Machines (RBMs)? We characterize the RBM's unnormalized log-likelihood function as a type of neural network, and through a series of simulation results relate these networks to ones whose representational properties are better understood. We show the surprising result that RBMs can efficiently capture any distribution whose density depends on the number of 1's in their input. We also provide the first known example of a particular type of distribution that provably cannot be efficiently represented by an RBM, assuming a realistic exponential upper bound on the weights. By formally demonstrating that a relatively simple distribution cannot be represented efficiently by an RBM our results provide a new rigorous justification for the use of potentially more expressive generative models, such as deeper ones.

## 1 Introduction

Standard Restricted Boltzmann Machines (RBMs) are a type of Markov Random Field (MRF) characterized by a bipartite dependency structure between a group of binary visible units $\mathbf{x} \in \{0,1\}^n$ and binary hidden units $h \in \{0,1\}^m$. Their energy function is given by:

$$E_\theta(\mathbf{x}, \mathbf{h}) = -\mathbf{x}^\top W \mathbf{h} - \mathbf{c}^\top \mathbf{x} - \mathbf{b}^\top \mathbf{h}$$

where $W \in \mathbb{R}^{n \times m}$ is the matrix of weights, $\mathbf{c} \in \mathbb{R}^n$ and $\mathbf{b} \in \mathbb{R}^m$ are vectors that store the input and hidden biases (respectively) and together these are referred to as the RBM's parameters $\theta = \{W, \mathbf{c}, \mathbf{b}\}$. The energy function specifies the probability distribution over the joint space $(\mathbf{x}, \mathbf{h})$ via the Boltzmann distribution $p(\mathbf{x}, \mathbf{h}) = \frac{1}{Z_\theta} \exp(-E_\theta(\mathbf{x}, \mathbf{h}))$ with the *partition function* $Z_\theta$ given by $\sum_{\mathbf{x},\mathbf{h}} \exp(-E_\theta(\mathbf{x}, \mathbf{h}))$. Based on this definition, the probability for any subset of variables can be obtained by conditioning and marginalization, although this can only be done efficiently up to a multiplicative constant due to the intractability of the RBM's partition function (Long and Servedio, 2010).

RBMs have been widely applied to various modeling tasks, both as generative models (e.g. Salakhutdinov and Murray, 2008; Hinton, 2000; Courville et al., 2011; Marlin et al., 2010; Tang and Sutskever, 2011), and for pre-training feed-forward neural nets in a layer-wise fashion (Hinton and Salakhutdinov, 2006). This method has led to many new applications in general machine learning problems including object recognition and dimensionality reduction. While promising for practical applications, the scope and basic properties of these statistical models have only begun to be studied.

As with any statistical model, it is important to understand the expressive power of RBMs, both to gain insight into the range of problems where they can be successfully applied, and to provide justification for the use of potentially more expressive generative models. In particular, we are interested in the question of how large the number of hidden units $m$ must be in order to capture a particular distribution to arbitrarily high accuracy. The question of size is of practical interest, since very large models will be computationally more demanding (or totally impractical), and will tend to overfit a lot more during training.

It was shown by Freund and Haussler (1994), and later by Le Roux and Bengio (2008) that for binary-valued $\mathbf{x}$, any distribution over $\mathbf{x}$ can be realized (up to an approximation error which vanishes exponentially quickly in the magnitude of the parameters) by an RBM, as long as $m$ is allowed to grow exponentially fast in input dimension ($n$). Intuitively, this construction works by instantiating, for each of the up to $2^n$ possible values of $\mathbf{x}$ that have support, a single hidden unit which turns on only for that particular value of $\mathbf{x}$ (with overwhelming probability), so that the corresponding probability mass can be individually set by manipulating that unit's bias parameter. An improvement to this result was obtained by Montufar and Ay (2011); however this construction still requires that $m$ grow exponentially fast in $n$.

Recently, Montufar et al. (2011) generalized the construction used by Le Roux and Bengio (2008) so that each hidden unit turns on for, and assigns probability mass to, not just a single $\mathbf{x}$, but a "cubical set" of possible $\mathbf{x}$'s, which is defined as a subset of $\{0,1\}^n$ where some entries of $\mathbf{x}$ are fixed/determined, and the rest are free. By combining such hidden units that are each specialized to a particular cubic set, they showed that any $k$-component mixture of product distributions over the free variables of mutually disjoint cubic sets can be approximated arbitrarily well by an RBM with $m = k$ hidden units.

Unfortunately, families of distributions that are of this specialized form (for some $m = k$ bounded by a polynomial function of $n$) constitute only a very limited subset of all distributions that have some kind of meaningful/interesting structure. For example, this result would not allow us to efficiently construct simple distributions where the mass is a function of $\sum_i x_i$ (e.g., for $p(\mathbf{x}) \propto \mathrm{PARITY}(\mathbf{x})$).

In terms of what kinds of distributions provably *cannot* be efficiently represented by RBMs, even less is known. Cueto et al. (2009) characterized the distributions that can be realized by a RBM with $k$ parameters as residing within a manifold inside the entire space of distributions on $\{0,1\}^n$ whose dimension depends on $k$. For sub-exponential $k$ this implies the existence of distributions which cannot be represented. However, this kind of result gives us no indication of what these hard-to-represent distributions might look like, leaving the possibility that they might all be structureless or otherwise uninteresting.

In this paper we first develop some tools and simulation results which relate RBMs to certain easier-to-analyze approximations, and to neural networks with 1 hidden layer of threshold units, for which many results about representational efficiency are already known (Maass, 1992; Maass et al., 1994; Hajnal et al., 1993). This opens the door to a range of potentially relevant complexity results, some of which we apply in this paper.

Next, we present a construction that shows how RBMs with $m = n^2 + 1$ can produce arbitrarily good approximations to any distribution where the mass is a *symmetric function* of the inputs (that is, it depends on $\sum_i \mathbf{x}_i$). One example of such a function is the (in)famous PARITY function, which was shown to be hard to compute in the perceptron model by the classic Minsky and Papert book from 1968. This distribution is highly non-smooth and has exponentially many modes.

Having ruled out distributions with symmetric mass functions as candidates for ones that are hard for RBMs to represent, we provide a concrete example of one whose mass computation involves only one additional operation vs computing PARITY, and yet whose reprentation by an RBM provably requires $m$ to grow exponentially with $n$ (assuming an exponental upper bound on the size of the RBM's weights). Because this distribution is particularly simple, it can be viewed as a special case of many other more complex types of distributions, and thus our results speak to the hardness of representing those distributions with RBMs as well.

Our results provide a fine delineation between what is "easy" for RBMs to represent, and what is "hard". Perhaps more importantly, they demonstrate that the distributions that cannot be efficiently represented by RBMs can have a relatively basic structure, and are not simply random in appearance as one might hope given the previous results. This provides perhaps the first completely rigorous justification for the use of deeper generative models such as Deep Boltzmann Machines (Salakhutdinov and Hinton, 2009), and contrastive backpropagation networks (Hinton et al., 2006) over standard RBMs.

The rest of the paper is organized as follows. Section 2 characterizes the unnormalized log-likelihood as a type of neural network (called an "RBM network") and shows how this type is related to single hidden layer neural networks of threshold neurons, and to an easier-to-analyze approximation (which we call a "hardplus RBM network"). Section 3 describes a $m = n^2 + 1$ construction for distributions whose mass is a function of $\sum x_i$, and in Section 4 we present an exponential lower bound on $m$ for a slightly more complicated class of explicit distributions. Note that all proofs can be found in the Appendix.

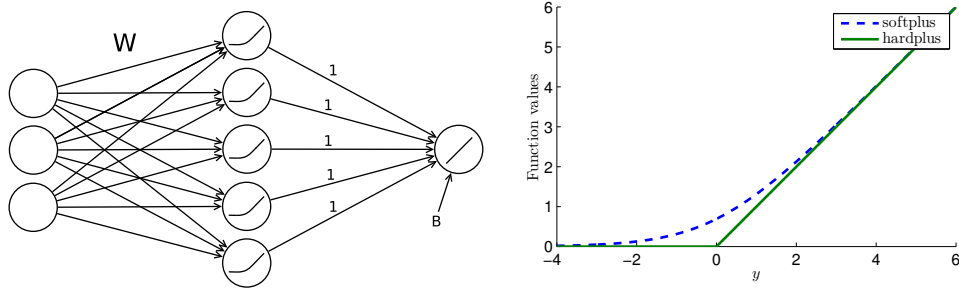

Figure 1: **Left:** An illustration of a basic RBM network with $n = 3$ and $m = 5$. The hidden biases are omitted to avoid clutter. **Right:** A plot comparing the soft and hard activation functions.

## 2 RBM networks

### 2.1 Free energy function

In an RBM, the (negative) unnormalized log probability of $\mathbf{x}$, after $\mathbf{h}$ has been marginalized out, is known as the *free energy*. Denoted by $F_\theta(\mathbf{x})$, the free energy satisfies the property that $p(\mathbf{x}) = \exp(-F_\theta(\mathbf{x}))/Z_\theta$ where $Z_\theta$ is the usual partition function.

It is well known (see Appendix A.1 for a derivation) that, due to the bipartite structure of RBMs, computing $F$ is tractable and has a particularly nice form:

$$F_\theta(\mathbf{x}) = -\mathbf{c}^\top \mathbf{x} - \sum_j \log(1 + \exp(\mathbf{x}^\top [W]_j + b_j)) \tag{1}$$

where $[W]_j$ is the $j$-th column of $W$.

Because the free energy completely determines the log probability of $\mathbf{x}$, it fully characterizes an RBM's distribution. So studying what kinds of distributions an RBM can represent amounts to studying the kinds of functions that can be realized by the free energy function for some setting of $\theta$.

### 2.2 RBM networks

The form of an RBM's free energy function can be expressed as a standard feed-forward neural network, or equivalently, a real-valued circuit, where instead of using hidden units with the usual sigmoidal activation functions, we have $m$ "neurons" (a term we will use to avoid confusion with the original meaning of a "unit" in the context of RBMs) that use the *softplus* activation function:

$$\text{soft}(y) = \log(1 + \exp(y))$$

Note that at the cost of increasing $m$ by one (which does not matter asymptotically) and introducing an arbitrarily small approximation error, we can assume that the visible biases ($\mathbf{c}$) of an RBM are all zero. To see this, note that up to an additive constant, we can very closely approximate $\mathbf{c}^\top \mathbf{x}$ by $\text{soft}(K + \mathbf{c}^\top \mathbf{x}) \approx K + \mathbf{c}^\top \mathbf{x}$ for a suitably large value of $K$ (i.e., $K \gg \|\mathbf{c}\|_1 \geq \max_{\mathbf{x}}(\mathbf{c}^\top \mathbf{x})$). Proposition 11 in the Appendix quantifies the very rapid convergence of this approximation as $K$ increases.

These observations motivate the following definition of an *RBM network*, which computes functions with the same form as the *negative* free energy function of an RBM (assumed to have $\mathbf{c} = 0$), or equivalently the log probability (negative energy) function of an RBM. RBM networks are illustrated in Figure 1.

**Definition 1** A RBM network with parameters $W, \mathbf{b}$ is defined as a neural network with one hidden layer containing $m$ softplus neurons and weights and biases given by $W$ and $\mathbf{b}$, so that each neuron $j$'s output is $\text{soft}([W]_j + b_j)$. The output layer contains one neuron whose weights and bias are given by $\mathbf{1} \equiv [11...1]^\top$ and the scalar $B$, respectively.

For convenience, we include the bias constant $B$ so that RBM networks shift their output by an additive constant (which does not affect the probability distribution implied by the RBM network since any additive constant is canceled out by $\log Z$ in the full log probability).

## 2.3 Hardplus RBM networks

A function which is somewhat easier to analyze than the softplus function is the so-called *hardplus* function (aka 'plus' or 'rectification'), defined by:

$$\text{hard}(y) = \max(0, y)$$

As their names suggest, the softplus function can be viewed as a smooth approximation of the hardplus, as illustrated in Figure 1. We define a *hardplus RBM network* in the obvious way: as an RBM network with the softplus activation functions of the hidden neurons replaced with hardplus functions.

The strategy we use to prove many of the results in this paper is to first establish them for hardplus RBM networks, and then show how they can be adapted to the standard softplus case via simulation results given in the following section.

## 2.4 Hardplus RBM networks versus (Softplus) RBM networks

In this section we present some approximate simulation results which relate hardplus and standard (softplus) RBM networks.

The first result formalizes the simple observation that for large input magnitudes, the softplus and hardplus functions behave very similarly (see Figure 1, and Proposition 11 in the Appendix).

**Lemma 2.** *Suppose we have a softplus and hardplus RBM networks with identical sizes and parameters. If, for each possible input $x \in \{0, 1\}^n$, the magnitude of the input to each neuron is bounded from below by $C$, then the two networks compute the same real-valued function, up to an error (measured by $|\cdot|$) which is bounded by $m \exp(-C)$.*

The next result demonstrates how to approximately simulate a RBM network with a hardplus RBM network while incurring an approximation error which shrinks as the number of neurons increases. The basic idea is to simulate individual softplus neurons with groups of hardplus neurons that compute what amounts to a piece-wise linear approximation of the smooth region of a softplus function.

**Theorem 3.** *Suppose we have a (softplus) RBM network with $m$ hidden neurons with parameters bounded in magnitude by $C$. Let $p > 0$. Then there exists a hardplus RBM network with $\leq 2m^2 p \log(mp) + m$ hidden neurons and with parameters bounded in magnitude by $C$ which computes the same function, up to an approximation error of $1/p$.*

Note that if $p$ and $m$ are polynomial functions of $n$, then the simulation produces hardplus RBM networks whose size is also polynomial in $n$.

## 2.5 Thresholded Networks and Boolean Functions

Many relevant results and proof techniques concerning the properties of neural networks focus on the case where the output is thresholded to compute a Boolean function (i.e. a binary classification). In this section we define some key concepts regarding output thresholding, and present some basic propositions that demonstrate how hardness results for computing Boolean functions via thresholding yield analogous hardness results for computing certain real-valued functions.

We say that a real-valued function $g$ *represents* a Boolean function $f$ with *margin* $\delta$ if for all $\mathbf{x}$ $g$ satisfies $|g(\mathbf{x})| \geq \delta$ and $\text{thresh}(g(\mathbf{x})) = f(\mathbf{x})$, where $\text{thresh}$ is the $0/1$ valued threshold function defined by:

$$\text{thresh}(a) = \left\{ \begin{array}{ll} 1 & : a \geq 0 \\ 0 & : a < 0 \end{array} \right.$$

We define a *thresholded neural network* (a distinct concept from a "threshold network", which is a neural network with hidden neurons whose activation function is $\text{thresh}$) to be a neural network whose output is a single real value, which is followed by an application of the threshold function. Such a network will be said to compute a given Boolean function $f$ with margin $\delta$ (similar to the concept of "separation" from Maass et al. (1994)) if the real valued input $g$ to the final threshold represents $f$ according to the above definition.

While the output of a thresholded RBM network does not correspond to the log probability of an RBM, the following observation spells out how we can use thresholded RBM networks to establish lower bounds on the size of an RBM network required to compute certain simple functions (i.e., real-valued functions that represent certain Boolean functions):

**Proposition 4.** *If an RBN network of size $m$ can compute a real-valued function $g$ which represents $f$ with margin $\delta$, then there exists a thresholded RBM network that computes $f$ with margin $\delta$.*

This statement clearly holds if we replace each instance of "RBM network" with "hardplus RBM network" above.

Using Theorem 3 we can prove a more interesting result which states that any lower bound result for thresholded *hardplus* RBMs implies a somewhat weaker lower bound result for *standard* RBM networks:

**Proposition 5.** *If an RBM network of size $\leq m$ with parameters bounded in magnitude by $C$ computes a function which represents a Boolean function $f$ with margin $\delta$, then there exists a thresholded hardplus RBM network of size $\leq 4m^2 \log(2m/\delta)/\delta + m$ with parameters bounded in magnitude by $C$ ($C$ can be $\infty$) that computes $f(\mathbf{x})$ with margin $\delta/2$*

This proposition implies that any exponential lower bound on the size of a thresholded hardplus RBM network will yield an exponential lower bound for (softplus) RBM networks that compute functions of the given form, provided that the margin $\delta$ is bounded from below by some function of the form $1/poly(n)$.

Intuitively, if $f$ is a Boolean function and no RBM network of size $m$ can compute a real-valued function that represents $f$ (with a margin $\delta$), this means that no RBM of size $m$ can represent any distribution where the log probability of each member of $\{\mathbf{x}|f(\mathbf{x}) = 1\}$ is at least $2\delta$ higher than each member of $\{\mathbf{x}|f(\mathbf{x}) = 0\}$. In other words, RBMs of this size cannot generate any distribution where the two "classes" implied by $f$ are separated in log probability by more than $2\delta$.

## 2.6 RBM networks versus standard neural networks

Viewing the RBM log probability function through the formalism of neural networks (or real-valued circuits) allows us to make use of known results for general neural networks, and helps highlight important differences between what an RBM can effectively "compute" (via its log probability) and what a standard neural network can compute.

There is a rich literature studying the complexity of various forms of neural networks, with diverse classes of activation functions, e.g., Maass (1992); Maass et al. (1994); Hajnal et al. (1993). RBM networks are distinguished from these, primarily because they have a single hidden layer and because the upper level weights are constrained to be $\mathbf{1}$.

For some activation functions this restriction may not be significant, but for soft/hard-plus neurons, whose output is always positive, it makes particular computations much more awkward (or perhaps impossible) to express efficiently. Intuitively, the $j^{th}$ softplus neuron acts as a "feature detector", which when "activated" by an input s.t. $\mathbf{x}^\top \mathbf{w}_j + b_j \gg 0$, can only contribute positively to the log probability of $\mathbf{x}$, according to an (asymptotically) affine function of $\mathbf{x}$ given by that neuron's input. For example, it is easy to design an RBM network that can (approximately) output 1 for input $\mathbf{x} = \vec{0}$ and 0 otherwise (i.e., have a single hidden neuron with weights $-M\mathbf{1}$ for a large $M$ and bias $b$ such that $\text{soft}(b) = 1$), but it is not immediately obvious how an RBM network could efficiently compute (or approximate) the function which is 1 on all inputs *except* $\mathbf{x} = \vec{0}$, and 0 otherwise (it turns out that a non-obvious construction exists for $m = n$). By comparison, standard threshold networks only requires 1 hidden neuron to compute such a function.

In fact, it is easy to show[1] that without the constraint on upper level weights, an RBM network would be, up to a linear factor, at least as efficient at representing real-valued functions as a neural network with 1 hidden layer of threshold neurons. From this, and from Theorem 4.1 of Maass et al. (1994), it follows that a thresholded RBM network is, up to a polynomial increase in size, at least as efficient at computing Boolean functions as 1-hidden layer neural networks with *any* "sigmoid-like" activation function[2], and polynomially bounded weights.

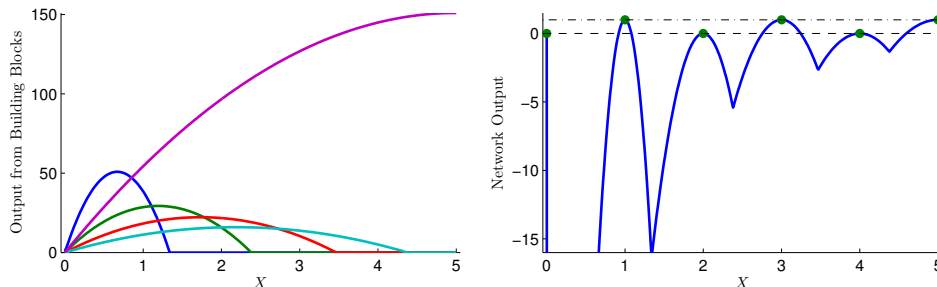

Figure 2: **Left:** The functions computed by the 5 building-blocks as constructed by Theorem 7 when applied to the PARITY function for $n = 5$. **Right:** The output total of the hardplus RBM network constructed in Theorem 7. The dotted lines indicate the target 0 and 1 values. **Note:** For purposes of illustration we have extended the function outputs over all real-values of $X$ in the obvious way.

## 2.7 Simulating hardplus RBM networks by a one-hidden-layer threshold network

Here we provide a natural simulation of hardplus RBM networks by threshold networks with one hidden layer. Because this is an efficient (polynomial) and exact simulation, it implies that a hardplus RBM network can be no more powerful than a threshold network with one hidden layer, for which several lower bound results are already known.

**Theorem 6.** *Let $f$ be a real-valued function computed by a hardplus RBM network of size $m$. Then $f$ can be computed by a single hidden layer threshold network, of size $mn$. Furthermore, if the weights of the RBM network have magnitude at most $C$, then the weights of the corresponding threshold network have magnitude at most $(n+1)C$.*

## 3 $n^2 + 1$-sized RBM networks can compute any symmetric function

In this section we present perhaps the most surprising results of this paper: a construction of an $n^2$-sized RBM network (or hardplus RBM network) for computing any given symmetric function of $\mathbf{x}$. Here, a *symmetric function* is defined as any real-valued function whose output depends only on the number of 1-bits in the input $\mathbf{x}$. This quantity is denoted $X \equiv \sum_i x_i$. A well-known example of a symmetric function is PARITY.

Symmetric functions are already known[3] to be computable by single hidden layer threshold networks (Hajnal et al., 1993) with $m = n$. Meanwhile (qualified) exponential lower bounds on $m$ exist for functions which are only slightly more complicated (Hajnal et al., 1993; Forster, 2002).

Given that hardplus RBM networks appear to be strictly less expressive than such threshold networks (as discussed in Section 2.6), it is surprising that they can nonetheless efficiently compute functions that test the limits of what those networks can compute efficiently.

**Theorem 7.** *Let $f : \{0,1\}^n \to \mathbb{R}$ be a symmetric function defined by $f(\mathbf{x}) = t_k$ for $\sum_i x_i = k$. Then (i) there exists a hardplus RBM network, of size $n^2 + 1$, and with weights polynomial in $n$ and $t_1, \ldots, t_k$ that computes $f$ exactly, and (ii) for every $\epsilon$ there is a softplus RBM network of size $n^2 + 1$, and with weights polynomial in $n$, $t_0, \ldots, t_n$ and $\log(1/\epsilon)$ that computes $f$ within an additive error $\epsilon$.*

The high level idea of this construction is as follows. Our hardplus RBM network consists of $n$ "building blocks", each composed of $n$ hardplus neurons, plus one additional hardplus neuron, for a total size of $m = n^2 + 1$. Each of these building blocks is designed to compute a function of the form:

$$\max(0, \gamma X(e - X))$$

for parameters $\gamma > 0$ and $e > 0$. This function, examples of which are illustrated in Figure 2, is quadratic from $X = 0$ to $X = e$ and is 0 otherwise.

The main technical challenge is then to choose the parameters of these building blocks so that the sum of $n$ of these "rectified quadratics", plus the output of the extra hardplus neuron (which handles

the $X = 0$ case), yields a function that matches $f$, up to a additive constant (which we then fix by setting the bias $B$ of the output neuron). This would be easy if we could compute more general rectified quadratics of the form $\max(0, \gamma(X - g)(e - X))$, since we could just take $g = k - 1/2$ and $e = k + 1/2$ for each possible value $k$ of $X$. But the requirement that $g = 0$ makes this more difficult since significant overlap between non-zero regions of these functions will be unavoidable. Further complicating the situation is the fact that we cannot exploit linear cancelations due to the restriction on the RBM network's second layer weights. Figure 2 depicts an example of the solution to this problem as given in our proof of Theorem 7.

Note that this construction is considerably more complex than the well-known construction used for computing symmetric functions with 1 hidden layer threshold networks Hajnal et al. (1993). While we cannot prove that ours is the most efficient possible construction RBM networks, we can prove that a construction directly analogous to the one used for 1 hidden layer threshold networks—where each individual neuron computes a symmetric function—cannot possibly work for RBM networks.

To see this, first observe that any neuron that computes a symmetric function must compute a function of the form $g(\beta X + b)$, where $g$ is the activation function and $\beta$ is some scalar. Then noting that both $\mathrm{soft}(y)$ and $\mathrm{hard}(y)$ are convex functions of $y$, and that the composition of an affine function and a convex function is convex, we have that each neuron computes a convex function of $X$. Then because the positive sum of convex functions is convex, the output of the RBM network (which is the unweighted sum of the output of its neurons, plus a constant) is itself convex in $X$. Thus the symmetric functions computable by such RBM networks must be convex in $X$, a severe restriction which rules out most examples.

# 4 Lower bounds on the size of RBM networks for certain functions

## 4.1 Existential results

In this section we prove a result which establishes the existence of functions which cannot be computed by RBM networks that are not exponentially large.

Instead of identifying non-representable distributions as lying in the complement of some low-dimensional manifold (as was done previously), we will establish the existence of Boolean functions which cannot be represented with a sufficiently large margin by the output of any sub-exponentially large RBM network. However, this result, like previous such existential results, will say nothing about what these Boolean functions actually look like.

To prove this result, we will make use of Proposition 5 and a classical result of Muroga (1971) which allows us to discretize the incoming weights of a threshold neuron (without changing the function it computes), thus allowing us to bound the number of possible Boolean functions computable by 1-layer threshold networks of size $m$.

**Theorem 8.** *Let $F_{m,\delta,n}$ represent the set of those Boolean functions on $\{0,1\}^n$ that can be computed by a thresholded RBM network of size $m$ with margin $\delta$. Then, there exists a fixed number $K$ such that,*

$$\left| F_{m,\delta,n} \right| \leq 2^{\mathrm{poly}(s,m,n,\delta)}, \quad where \quad s(m,\delta,n) = \frac{4m^2 n}{\delta} \log\left(\frac{2m}{\delta}\right) + m.$$

*In particular, when $m^2 \leq \delta 2^{\alpha n}$, for any constant $\alpha < 1/2$, the ratio of the size of the set $F_{m,\delta,n}$ to the total number of Boolean functions on $\{0,1\}^n$ (which is $2^{2^n}$), rapidly converges to zero with $n$.*

## 4.2 Qualified lower bound results for the IP function

While interesting, existential results such as the one above does not give us a clear picture of what a particular hard-to-compute function for RBM networks might look like. Perhaps these functions will resemble purely random maps without any interesting structure. Perhaps they will consist only of functions that require exponential time to compute on a Turing machine, or even worse, ones that are non-computable. In such cases, not being able to compute such functions would not constitute a meaningful limitation on the expressive efficiency of RBM networks.

In this sub-section we present strong evidence that this is not the case by exhibiting a simple Boolean function that provably requires exponentially many neurons to be computed by a thresholded RBM network, provided that the margin is not allowed to be exponentially smaller than the weights. Prior to these results, there was no formal separation between the kinds of unnormalized log-likelihoods realizable by polynomially sized RBMs, and the class of functions computable efficiently by almost any reasonable model of computation, such as arbitrarily deep Boolean circuits.

The Boolean function we will consider is the well-known "inner product mod 2" function, denoted $IP(\mathbf{x})$, which is defined as the parity of the the inner product of the first half of $\mathbf{x}$ with the second half (we assume for convenience that $n$ is even). This function can be thought of as a strictly harder to compute version of PARITY (since PARITY is trivially reducible to it), which as we saw in Section 7, can be efficiently computed by thresholded RBM network (indeed, an RBM network can efficiently compute any possible real-valued representation of PARITY). Intuitively, $IP(\mathbf{x})$ should be harder than PARITY, since it involves an extra "stage" or "layer" of sequential computation, and our formal results with RBMs agree with this intuition.

There are many computational problems that $IP$ can be reduced to, so showing that RBM networks cannot compute $IP$ thus proves that RBMs cannot efficiently model a wide range of distributions whose unnormalized log-likelihoods are sufficiently complex in a computational sense. Examples of such log-likelihoods include ones given by the multiplication of binary-represented integers, or the evaluation of the connectivity of an encoded graph. For other examples, see see Corollary 3.5 of Hajnal et al. (1993).

Using the simulation of hardplus RBM networks by 1 hidden layer threshold networks (Theorem 6), and Proposition 5, and an existing result about the hardness of computing $IP$ by 1 hidden layer thresholded networks of bounded weights due to Hajnal et al. (1993), we can prove the following basic result:

**Theorem 9.** *If* $m < \min \left\{ \frac{2^{n/3}}{C}, \ 2^{n/6} \sqrt{\frac{\delta}{4C \log(2/\delta)}}, \ 2^{n/9} \sqrt[3]{\frac{\delta}{4C}} \right\}$ *then no RBM network of size $m$, whose weights are bounded in magnitude by $C$, can compute a function which represents $n$-dimensional $IP$ with margin $\delta$. In particular, for $C$ and $1/\delta$ bounded by polynomials in $n$, for $n$ sufficiently large, this condition is satisfied whenever $m < 2^{(1/9-\epsilon)n}$ for some $\epsilon > 0$.*

Translating the definitions, this results says the following about the limitations of efficient representation by RBMs: Unless either the weights, or the number units of an RBM are exponentially large in $n$, an RBM cannot capture any distribution that has the property that $\mathbf{x}$'s s.t. IP($\mathbf{x}$) = 1 are significantly more probable than the remaining $\mathbf{x}$'s.

While the above theorem is easy to prove from known results and the simulation/hardness results given in previous sections, by generalizing the techniques used in Hajnal et al. (1993), we can (with much more effort) derive a stronger result. This gives an improved bound on $m$ and lets us partially relax the magnitude bound on parameters so that they can be arbitrarily *negative*:

**Theorem 10.** *If* $m < \frac{\delta}{2 \cdot max\{\log 2, nC + \log 2\}} \cdot 2^{n/4}$, *then no RBM network of size $m$, whose weights are upper bounded in* value *by $C$, can compute a function which represents $n$-dimensional $IP$ with margin $\delta$. In particular, for $C$ and $1/\delta$ bounded by polynomials in $n$, for $n$ sufficiently large, this condition is satisfied whenever $m < 2^{(1/4-\epsilon)n}$ for some $\epsilon > 0$.*

The general theorem we use to prove this second result (Theorem 17 in the Appendix) requires only that the neural network have 1 hidden layer of neurons with activation functions that are monotonic and contribute to the top neuron (after multiplication by the outgoing weight) a quantity which can be bounded by a certain exponentially growing function of $n$ (that also depends on $\delta$). Thus this technique can be applied to produce lower bounds for much more general types of neural networks, and thus may be independently interesting.

## 5 Conclusions and Future Work

In this paper we significantly advanced the theoretical understanding of the representational efficiency of RBMs. We treated the RBM's unnormalized log likelihood as a neural network which allowed us to relate an RBM's representational efficiency to that of threshold networks, which are much better understood. We showed that, quite suprisingly, RBMs can efficiently represent distributions that are given by symmetric functions such as PARITY, but cannot efficiently represent distributions which are slightly more complicated, assuming an exponential bound on the weights. This provides rigorous justification for the use of potentially more expressive/deeper generative models.

Going forward, some promising research directions and open problems include characterizing the expressive power of Deep Boltzmann Machines and more general Boltzmann machines, and proving an exponential lower bound for some specific distribution without any qualifications on the weights.

**Acknowledgments**
This research was supported by NSERC. JM is supported by a Google Fellowship; AC by a Ramanujan Fellowship of the DST, India.

## Footnotes

[1]To see this, note that we could use 2 softplus neurons to simulate a single neuron with a "sigmoid-like" activation function (i.e., by setting the weights that connect them to the output neuron to have opposite signs). Then, by increasing the size of the weights so the sigmoid saturates in both directions for all inputs, we could simulate a threshold function arbitrarily well, thus allowing the network to compute any function computable by a one hidden layer threshold network while only using only twice as many neurons.

[2]This is a broad class and includes the standard logistic sigmoid. See Maass et al. (1994) for a precise technical definition

[3]The construction in Hajnal et al. (1993) is only given for Boolean-valued symmetric functions but can be generalized easily.

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

# A Appendix

## A.1 Free-energy derivation

The following is a derivation of the well-known formula for the free-energy of an RBM. This tractable form is made possible by the bipartite interaction structure of the RBM's units:

$$
\begin{aligned}
p(\mathbf{x}) &= \sum_{\mathbf{h}} \frac{1}{Z_\theta} \exp(\mathbf{x}^\top W \mathbf{h} + \mathbf{c}^\top \mathbf{x} + \mathbf{b}^\top \mathbf{h}) \\
&= \frac{1}{Z_\theta} \exp(\mathbf{c}^\top \mathbf{x}) \prod_j \sum_{h_j \in \{0,1\}} \exp(\mathbf{x}^\top [W]_j h_j + b_j h_j) \\
&= \frac{1}{Z_\theta} \exp(\mathbf{c}^\top \mathbf{x}) \exp(\sum_j (\log \sum_{h_j \in \{0,1\}} \exp(\mathbf{x}^\top [W]_j h_j + b_j h_j))) \\
&= \frac{1}{Z_\theta} \exp(\mathbf{c}^\top \mathbf{x} + \sum_j \log[1 + \exp(\mathbf{x}^\top [W]_j + b_j)]) \\
&= \frac{1}{Z_\theta} \exp(-F_\theta(\mathbf{x}))
\end{aligned}
$$

## A.2 Proofs for Section 2.4

We begin with a useful technical result:

**Proposition 11.** *For arbitrary $y \in \mathbb{R}$ the following basic facts for the softplus function hold:*

$$
y - \mathrm{soft}(y) = -\mathrm{soft}(-y)
$$
$$
\mathrm{soft}(y) \leq \exp(y)
$$

*Proof.* The first fact follows from:

$$
y - \mathrm{soft}(y) = \log(\exp(y)) - \log(1 + \exp(y)) = \log\left(\frac{\exp(y)}{1 + \exp(y)}\right)
$$
$$
= \log\left(\frac{1}{\exp(-y) + 1}\right) = -\log(1 + \exp(y)) = -\mathrm{soft}(-y)
$$

To prove the second fact, we will show that the function $f(y) = \exp(y) - \mathrm{soft}(y)$ is positive. Note that $f$ tends to 0 as $y$ goes to $-\infty$ since both $\exp(y)$ and $\mathrm{soft}(y)$ do. It remains to show that $f$ is monotonically increasing, which we establish by showing that its derivative is positive:

$$
f'(y) = \exp(y) - \frac{1}{1 + \exp(-y)} > 0
$$
$$
\Leftrightarrow \quad \exp(y)(1 + \exp(-y)) - \frac{1 + \exp(-y)}{1 + \exp(-y)} > 0
$$
$$
\Leftrightarrow \quad \exp(y) + 1 - 1 > 0 \quad \Leftrightarrow \quad \exp(y) > 0
$$

$\square$

**Proof of Lemma 2.** Consider a single neuron in the RBM network and the corresponding neuron in the hardplus RBM network, whose net-input are given by $y = w^\top x + b$.

For each $x$, there are two cases for $y$. If $y \geq 0$, we have by hypothesis that $y \geq C$, and so:

$$
|\mathrm{hard}(y) - \mathrm{soft}(y)| = |y - \mathrm{soft}(y)| = |-\mathrm{soft}(-y)| = \mathrm{soft}(-y)
$$
$$
\leq \exp(-y) \leq \exp(-C)
$$

And if $y < 0$, we have by hypothesis that $y \leq -C$ and so:

$$
|\mathrm{hard}(y) - \mathrm{soft}(y)| = |0 - \mathrm{soft}(y)| = \mathrm{soft}(y)
$$
$$
\leq \exp(y) \leq \exp(-C)
$$

Thus, each corresponding pair of neurons computes the same function up to an error bounded by $\exp(-C)$. From this it is easy to show that the entire circuits compute the same function, up to an error bounded by $m \exp(-C)$, as required. $\qquad \square$

**Proof of Theorem 3.** Suppose we have a softplus RBM network with a number of hidden neurons given by $m$. To simulate this with a hardplus RBM network we will replace each neuron with a group of hardplus neurons with weights and biases chosen so that the sum of their outputs approximates the output of the original softplus neuron, to within a maximum error of $1/p$ where $p$ is some constant $> 0$.

First we describe the construction for the simulation of a single softplus neurons by a group of hardplus neurons.

Let $g$ be a positive integer and $a > 0$. We will define these more precisely later, but for what follows their precise value is not important.

At a high level, this construction works by approximating $\text{soft}(y)$, where $y$ is the input to the neuron, by a piece-wise linear function expressed as the sum of a number of hardplus functions, whose "corners" all lie inside $[-a, a]$. Outside this range of values, we use the fact that $\text{soft}(y)$ converges exponentially fast (in $a$) to $0$ on the left, and $y$ on the right (which can both be trivially computed by hardplus functions).

Formally, for $i = 1, 2, ..., g, g + 1$, let:

$$q_i = (i - 1)\frac{2a}{g} - a$$

For $i = 1, 2, ..., g$, let:

$$\nu_i = \frac{\text{soft}(q_{i+1}) - \text{soft}(q_i)}{q_{i+1} - q_i}$$

and also let $\nu_0 = 0$ and $\nu_{g+1} = 1$. Finally, for $i = 1, 2, ..., g, g + 1$, let:

$$\eta_i = \nu_i - \nu_{i-1}$$

With these definitions it is straightforward to show that $1 \geq \nu_i > 0$, $\nu_i > \nu_{i-1}$ and consequently $0 < \eta_i < 1$ for each $i$. It is also easy to show that $q_i > q_{i-1}$, $q_0 = -a$ and $q_{g+1} = a$.

For $i = 1, 2, ..., g, g + 1$, we will set the weight vector $\mathbf{w}_i$ and bias $b_i$ of the $i$-th hardplus neuron in our group so that the neuron outputs $\text{hard}(\eta_i(y - q_i))$. This is accomplished by taking $\mathbf{w}_i = \eta_i \mathbf{w}$ and $b_i = \eta_i(b - q_i)$, where $\mathbf{w}$ and $b$ (without the subscripts), are the weight vector and bias of the original softplus neuron.

Note that since $|\eta_i| \leq 1$ we have that the weights of these $\text{hard}$ neurons are smaller in magnitude than the weights of the original $\text{soft}$ neuron and thus bounded by $C$ as required.

The total output (sum) for this group is:

$$T(y) = \sum_{i=1}^{g+1} \text{hard}(\eta_i(y - q_i))$$

We will now bound the approximation error $|T(y) - \text{soft}(y)|$ of our single neuron simulation.

Note that for a given $y$ we have that the $i$-th hardplus neuron in the group has a non-negative input iff $y \geq q_i$. Thus for $y < -a$ all of the neurons have a negative input. And for $y \geq -a$, if we take $j$ to be the largest index $i$ s.t. $q_i \leq y$, then each neuron from $i = 1$ to $i = j$ will have positive input and each neuron from $i = j + 1$ to $i = g + 1$ will have negative input.

Consider the case that $y < -a$. Since the input to each neuron is negative, they each output $0$ and thus $T(y) = 0$. This results in an approximation error $\leq \exp(-a)$:

$$|T(y) - \text{soft}(y)| = |0 - \text{soft}(y)| = \text{soft}(y) < \text{soft}(-a) \leq \exp(-a)$$

Next, consider the case that $y \geq -a$, and let $j$ be as given above. In such a case we have:

$$T(y) = \sum_{i=1}^{g+1} \text{hard}(\eta_i(y - q_i)) = \sum_{i=1}^{j} \eta_i(y - q_i) + 0$$

$$= \sum_{i=1}^{j} (\nu_i - \nu_{i-1})(y - q_i)$$

$$= y \sum_{i=1}^{j} (\nu_i - \nu_{i-1}) - \sum_{i=1}^{j} (\nu_i - \nu_{i-1}) q_i$$

$$= y\nu_j - y\nu_0 - \nu_j q_j + \sum_{i=1}^{j-1} \nu_i(q_{i+1} - q_i) + \nu_0 q_1$$

$$= \nu_j(y - q_j) + \sum_{i=1}^{j-1} (\text{soft}(q_{i+1}) - \text{soft}(q_i))$$

$$= \nu_j(y - q_j) + \text{soft}(q_j) - \text{soft}(q_1)$$

For $y \leq a$ we note that $\nu_j(y - q_j) + \text{soft}(q_j)$ is a secant approximation to $\text{soft}(y)$ generated by the secant from $q_j$ to $q_{j+1}$ and upperbounds $\text{soft}(y)$ for $y \in [q_j, q_{j+1}]$. Thus a crude bound on the error is $\text{soft}(q_{j+1}) - \text{soft}(q_j)$, which only makes use of the fact that $\text{soft}(y)$ is monotonic. Then because the slope (derivative) of $\text{soft}(y)$ is $\sigma(y) = 1/(1 + \exp(-y)) < 1$, we can further (crudely) bound this by $q_{j+1} - q_j$. Thus the approximation error at such $y$'s may be bounded as:

$$|T(y) - \text{soft}(y)| = |(\nu_j(y - q_j) + \text{soft}(q_j) - \text{soft}(q_1)) - \text{soft}(y)|$$

$$\leq \max\{|\nu_j(y - q_j) + \text{soft}(q_j) - \text{soft}(y)|, \text{soft}(q_1)\}$$

$$\leq \max\{q_{j+1} - q_j, \exp(-a)\} = \max\left\{\frac{2a}{g}, \exp(-a)\right\}$$

where we have also used $\text{soft}(q_1) = \text{soft}(-a) \leq \exp(-a)$.

For the case $y > a$, all $q_i > y$ and the largest index $j$ such that $q_j \leq y$ is $j = g + 1$. So $\nu_j(y - q_j) + \text{soft}(q_j) - \text{soft}(q_1) = y - a + \text{soft}(a) - \text{soft}(-a) = y$. Thus the approximation error at such $y$'s is:

$$|y - \text{soft}(y)| = |-\text{soft}(-y)| = \text{soft}(-y) \leq \text{soft}(-a) \leq \exp(-a)$$

Having covered all cases for $y$ we conclude that the general approximation error for a single softplus neuron satisfies the following bound:

$$|y - \text{soft}(y)| \leq \max\left\{\frac{2a}{g}, \exp(-a)\right\}$$

For a softplus RBM network with $m$ neurons, our hardplus RBM neurons constructed by replacing each neuron with a group of hardplus neurons as described above will require a total of $m(g + 1)$ neurons, and have an approximation error bounded by the sum of the individual approximation errors, which is itself bounded by:

$$m \max\left\{\frac{2a}{g}, \exp(-a)\right\}$$

Taking $a = \log(mp)$, $g = \lceil 2mpa \rceil$. This gives:

$$m \max\left\{\frac{2a}{\lceil 2mpa \rceil}, \frac{1}{mp}\right\} \leq m \max\left\{\frac{2a}{2mpa}, \frac{1}{mp}\right\}$$

$$= \max\left\{\frac{1}{p}, \frac{1}{p}\right\} = \frac{1}{p}$$

Thus we see that with $m(g + 1) = m(\lceil 2mp\log(mp) \rceil + 1) \leq 2m^2p\log(mp) + m$ neurons we can produce a hardplus RBM network which approximates the output of our softplus RBM network with error bounded by $1/p$. □

**Remark 12.** *Note that the construction used in the above lemma is likely far from optimal, as the placement of the $q_i$'s could be done more carefully. Also, the error bound we proved is crude and does not make strong use of the properties of the softplus function. Nonetheless, it seems good enough for our purposes.*

## A.3 Proofs for Section 2.5

*Proof of Proposition 5.* Suppose that there is an RBM network of size $m$ with weights bounded in magnitude by $C$ computes a function $g$ which represent $f$ with margin $\delta$.

Then taking $p = 2/\delta$ and applying Theorem 3 we have that there exists an hardplus RBM network of size $4m^2 \log(2m/\delta)/\delta + m$ which computes a function $g'$ s.t. $|g(\mathbf{x}) - g'(\mathbf{x})| \leq 1/p = \delta/2$ for all $\mathbf{x}$.

Note that $f(\mathbf{x}) = 1 \Rightarrow \mathrm{thresh}(g(\mathbf{x})) = 1 \Rightarrow g(\mathbf{x}) \geq \delta \Rightarrow g'(\mathbf{x}) \geq \delta - \delta/2 = \delta/2$ and similarly, $f(\mathbf{x}) = 0 \Rightarrow \mathrm{thresh}(g(\mathbf{x})) = 0 \Rightarrow g(\mathbf{x}) \leq -\delta \Rightarrow g'(\mathbf{x}) \leq -\delta + \delta/2 = -\delta/2$. Thus we conclude that $g'$ represents $f$ with margin $\delta/2$.

$\square$

## A.4 Proofs for Section 2.7

*Proof of Theorem 6.* Let $f$ be a Boolean function on $n$ variables computed by a size $s$ hardplus RBM network, with parameters $(W, b, d)$. We will first construct a three layer hybrid Boolean/threshold circuit/network where the output gate is a simple weighted sum, the middle layer consists of AND gates, and the bottom hidden layer consists of threshold neurons. There will be $n \cdot m$ AND gates, one for every $i \in [n]$ and $j \in [m]$. The $(i,j)^{th}$ AND gate will have inputs: (1) $x_i$ and (2) $(\mathbf{x}^\top [W]_j \geq b_j)$. The weights going from the $(i,j)^{th}$ AND gate to the output will be given by $[W]_{i,j}$. It is not hard to see that our three layer netork computes the same Boolean function as the original hardplus RBM network.

In order to obtain a single hidden layer threshold network, we replace each sub-network rooted at an AND gate of the middle layer by a single threshold neuron. Consider a general sub-network consisting of an AND of: (1) a variable $x_j$ and (2) a threshold neuron computing $(\sum_{i=1}^n a_i x_i \geq b)$. Let $Q$ be some number greater than the sum of all the $a_i$'s. We replace this sub-network by a single threshold gate that computes $(\sum_{i=1}^n a_i x_i + Q x_j \geq b + Q)$. Note that if the input $x$ is such that $\sum_i a_i x_i \geq b$ and $x_j = 1$, then $\sum_i a_i x_i + Q \alpha_j$ will be at least $b + Q$, so the threshold gate will output 1. In all other cases, the threshold will output zero. (If $\sum_i a_i x_i < b$, then even if $x_j = 1$, the sum will still be less than $Q + b$. Similarly, if $x_j = 0$, then since $\sum_i a_i x_i$ is never greater than $\sum_i a_i$, the total sum will be less than $Q \leq (n+1)C$.)

$\square$

## A.5 Proof of Theorem 7

*Proof.* We will first describe how to construct a hardplus RBM network which satisfies the properties required for part (i). It will be composed of $n$ special groups of hardplus neurons (which are defined and discussed below), and one additional one we call the "zero-neuron", which will be defined later.

**Definition 13** A "building block" is a group of $n$ hardplus neurons, parameterized by the scalars $\gamma$ and $e$, where the weight vector $\mathbf{w} \in \mathbb{R}^n$ between the $i$-th neuron in the group and the input layer is given by $w_i = M - \gamma$ and $w_j = -\gamma$ for $j \neq i$ and the bias will be given by $b = \gamma e - M$, where $M$ is a constant chosen so that $M > \gamma e$.

For a given $\mathbf{x}$, the input to the $i$-th neuron of a particular building block is given by:

$$\sum_{j=1}^n w_j x_j + b = w_i x_i + \sum_{j \neq i} w_j x_j + b$$
$$= (M - \gamma)x_i - \gamma(X - x_i) + \gamma e - M$$
$$= \gamma(e - X) - M(1 - x_i)$$

When $x_i = 0$, this is $\gamma(e - X) - M < 0$, and so the neuron will output 0 (by definition of the hardplus function). On the other hand, when $x_i = 1$, the input to the neuron will be $\gamma(e - X)$ and thus the output will be $\max(0, \gamma(e - X))$.

In general, we have that the output will be given by:

$$x_i \max(0, \gamma(e - X))$$

From this it follows that the combined output from the neurons in the building block is:

$$\sum_{i=1}^{n} (x_i \max(0, \gamma(e - X))) = \max(0, \gamma(e - X)) \sum_{i=1}^{n} x_i$$
$$= \max(0, \gamma(e - X))X = \max(0, \gamma X(e - X))$$

Note that whenever $X$ is positive, the output is a concave quadratic function in $X$, with zeros at $X = 0$ and $X = e$, and maximized at $X = e/2$, with value $\gamma e^2/4$.

Next we show how the parameters of the $n$ building blocks used in our construction can be set to produce a hardplus RBM network with the desired output.

First, define $d$ to be any number greater than or equal to $2n^2 \sum_j |t_j|$.

Indexing the building blocks by $j$ for $1 \leq j \leq n$ we define their respective parameters $\gamma_j$, $e_j$ as follows:

$$\gamma_n = \frac{t_n + d}{n^2}, \qquad \gamma_j = \frac{t_j + d}{j^2} - \frac{t_{j+1} + d}{(j+1)^2}$$
$$e_n = 2n, \qquad e_j = \frac{2}{\gamma_j}\left(\frac{t_j + d}{j} - \frac{t_{j+1} + d}{j+1}\right)$$

where we have assumed that $\gamma_j \neq 0$ (which will be established, along with some other properties of these definitions, in the next claim).

**Claim 1.** *For all $j$, $1 \leq j \leq n$, (i) $\gamma_j > 0$ and (ii) for all $j$, $1 \leq j \leq n - 1$, $j \leq e_j \leq j + 1$.*

*Proof of Claim 1.* Part (i): For $j = n$, by definition we know that $\gamma_n = \frac{t_n + d}{n^2}$. For $d \geq 2n^2 \sum_j |t_j| > |t_n|$, the numerator will be positive and therefore $\gamma_n$ will be positive.

For $j < n$, we have:

$$\gamma_j > 0$$
$$\Leftrightarrow \quad \frac{t_j + d}{j^2} > \frac{t_{j+1} + d}{(j+1)^2}$$
$$\Leftrightarrow \quad (j+1)^2(t_j + d) > j^2(t_{j+1} + d)$$
$$\Leftrightarrow \quad d((j+1)^2 - j^2) > j^2 t_{j+1} - (j+1)^2 t_j$$
$$\Leftrightarrow \quad d > \frac{j^2 t_{j+1} - (j+1)^2 t_j}{2j + 1}$$

The right side of the above inequality is less than or equal to $\frac{(j+1)^2(|t_{j+1}|+|t_j|)}{2j+1} \leq (j+1)(|t_{j+1}|+|t_j|)$ which is strictly upper bounded by $2n^2 \sum_j |t_j|$, and thus by $d$. So it follows that $\gamma_j > 0$ as needed.

Part (ii):

$$j \le e_j = \frac{2}{\gamma_j}\left(\frac{t_j+d}{j} - \frac{t_{j+1}+d}{j+1}\right)$$

$$\Leftrightarrow \quad j\gamma_j \le 2\left(\frac{t_j+d}{j} - \frac{t_{j+1}+d}{j+1}\right)$$

$$\Leftrightarrow \quad \frac{t_j+d}{j} - \frac{j(t_{j+1}+d)}{(j+1)^2} \le 2\left(\frac{t_j+d}{j} - \frac{t_{j+1}+d}{j+1}\right)$$

$$\Leftrightarrow \quad -\frac{j(t_{j+1}+d)}{(j+1)^2} \le \frac{t_j+d}{j} - 2\frac{t_{j+1}+d}{j+1}$$

$$\Leftrightarrow \quad -(t_{j+1}+d)j^2 \le (t_j+d)(j+1)^2 - 2(t_{j+1}+d)j(j+1)$$

$$\Leftrightarrow \quad d(j^2-2j(j+1)+(j+1)^2) \ge -j^2 t_{j+1} + 2j(j+1)t_{j+1} - (j+1)^2 t_j$$

$$\Leftrightarrow \quad d \ge -j^2 t_{j+1} + 2j(j+1)t_{j+1} - (j+1)^2 t_j$$

where we have used $j^2 - 2j(j+1) + (j+1)^2 = (j-(j+1))^2 = 1^2 = 1$ at the last line. Thus it suffices to make $d$ large enough to ensure that $j \le e_j$. For our choice of $d$, this will be true.

For the upper bound we have:

$$\frac{2}{\gamma_j}\left(\frac{t_j+d}{j} - \frac{t_{j+1}+d}{j+1}\right) = e_j \le j+1$$

$$\Leftrightarrow \quad 2\left(\frac{t_j+d}{j} - \frac{t_{j+1}+d}{j+1}\right) \le (j+1)\gamma_j = \frac{(j+1)(t_j+d)}{j^2} - \frac{t_{j+1}+d}{j+1}$$

$$\Leftrightarrow \quad 2\frac{t_j+d}{j} - \frac{t_{j+1}+d}{j+1} \le \frac{(j+1)(t_j+d)}{j^2}$$

$$\Leftrightarrow \quad 2(t_j+d)j(j+1) - (t_{j+1}+d)j^2 \le (j+1)^2(t_j+d)$$

$$\Leftrightarrow \quad \frac{-(d-t_{j+1})}{j+1} + 2\frac{(d+t_j)}{j} \le (j+1)\frac{(d+t_j)}{j^2}$$

$$\Leftrightarrow \quad -j^2(d+t_{j+1}) + 2j(j+1)(d+t_j) \le (j+1)^2(d+t_j)$$

$$\Leftrightarrow \quad d(j^2 - 2j(j+1) + (j+1)^2)$$
$$\ge -j^2 t_{j+1} + 2j(j+1)t_j - (j+1)^2 t_j$$

$$\Leftrightarrow \quad d \ge -j^2 t_{j+1} + 2j(j+1)t_j - (j+1)^2 t_j$$

where we have used $j^2 - 2j(j+1) + (j+1)^2 = 1$ at the last line. Again, for our choice of $d$ the above inequality is satisfied.

$\square$

Finally, define $M$ to be any number greater than $\max(t_0 + d, \max_i\{\gamma_i e_i\})$.

In addition to the $n$ building blocks, our hardplus RBM will include an addition unit that we will call the *zero*-neuron, which handles $\mathbf{x} = \mathbf{0}$. The zero-neuron will have weights $\mathbf{w}$ defined by $w_i = -M$ for each $i$, and $b = t_0 + d$.

Finally, the output bias $B$ of our hardplus RBM network will be set to $-d$.

The total output of the network is simply the sum of the outputs of the $n$ different building blocks, the zero neuron, and constant bias $-d$.

To show part (i) of the theorem we want to prove that for all $k$, whenever $X = k$, our circuit outputs the value $t_k$.

We make the following definitions:

$$a_k \equiv -\sum_{j=k}^{n} \gamma_j \qquad b_k \equiv \sum_{j=k}^{n} \gamma_j e_j$$

**Claim 2.**

$$a_k = \frac{-(t_k + d)}{k^2} \qquad b_k = \frac{2(t_k + d)}{k} \qquad b_k = -2ka_k$$

This claim is self-evidently true by examining basic definitions of $\gamma_j$ and $e_j$ and realizing that $a_k$ and $b_k$ are telescoping sums.

Given these facts, we can prove the following:

**Claim 3.** *For all $k$, $1 \leq k \leq n$, when $X = k$ the sum of the outputs of all the $n$ building blocks is given by $t_k + d$.*

*Proof of Claim 3.* For $X = n$, the $(\gamma_n, e_n)$-block computes $\max(0, \gamma_n X(e_n - X)) = \max(0, -\gamma_n X^2 + \gamma_n e_n X)$. By the definition of $e_n$, $n \leq e_n$, and thus when $X \leq n$, $\gamma_n X(e_n - X) \geq 0$. For all other building blocks $(\gamma_j, e_j)$, $j < n$, since $e_j \leq j + 1$, this block outputs zero since $\gamma_j X(e_j - X)$ is less than or equal to zero. Thus the sum of all of the building blocks when $X = n$ is just the output of the $(\gamma_n, e_n)$-block which is

$$\gamma_n \cdot n(e_n - n) = -\gamma_n \cdot n^2 + \gamma_n e_n \cdot n = -(t_n + d) + 2(t_n + d) = t_n + d$$

as desired.

For $X = k$, $1 \leq k < n$ the argument is similar. For all building blocks $j \geq k$, by Claim 1 we know that $e_j \geq j$ and therefore this block on $X = k$ is nonnegative and therefore contributes to the sum. On the other hand, for all building blocks $j < k$, by Claim 1 we know that $e_j \leq j + 1$ and therefore this outputs 0 and so does not contribute to the sum.

Thus the sum of all of the building blocks is equal to the sum of the non-zero regions of the building blocks $j$ for $j \geq k$. Since each of this is a quadratic function of $X$, it can written as a single quadratic polynomial of the form $a_k X^2 + b_k X$ where $a_k$ and $b_k$ are defined as before.

Plugging in the above expressions for $a_k$ and $b_k$ from Claim 2, we see that the value of this polynomial at $X = k$ is:

$$a_k k^2 + b_k k = \frac{-(t_k + d)}{k^2} k^2 + \frac{2(t_k + d)}{k} k = -(t_k + d) + 2(t_k + d) = t_k + d$$

$\square$

Finally, it remains to ensure that our hardplus RBM network outputs $t_0$ for $X = 0$. Note that the sum of the outputs of all $n$ building blocks and the output bias is $-d$ at $X = 0$. To correct this, we set the incoming weights and the bias of the zero-neuron according to $w_i = -M$ for each $i$, and $b = t_0 + d$. When $X = 0$, this neuron will output $t_0 + d$, making the total output of the network $-d + t_0 + d = t_0$ as needed. Furthermore, note that the addition of the zero-neuron does not affect the output of the network when $X = k > 0$ because the zero-neuron outputs 0 on all of these inputs as long as $M \geq t_0 + d$.

This completes the proof of part (i) of the theorem and it remains to prove part (ii).

Observe that the size of the weights grows linearly in $M$ and $d$, which follows directly from their definitions. And note that the magnitude of the input to each neuron is lower bounded by a positive linear function of $M$ and $d$ (a non-trivial fact which we will prove below). From these two observations it follows that to achieve the condition that the magnitude of the input to each neuron is greater than $C(n)$ for some function $C$ of $n$, the weights need to grow linearly with $C$. Noting that error bound condition $\epsilon \leq (n^2 + 1) \exp(-C)$ in Lemma 2 can be rewritten as $C \leq \log((n^2 + 1)) + \log(1/\epsilon)$, from which part (ii) of the theorem then follows.

There are two cases where a hardplus neuron in building block $j$ has a negative input. Either the input is $\gamma_j(e_j - X) - M$, or it is $\gamma_j(e_j - X)$ for $X \geq j + 1$. In the first case it is clear that as $M$ grows the net input becomes more negative since $e_j$ doesn't depend on $M$ at all.

The second case requires more work. First note that from its defintion, $e_j$ can be rewritten as $2\frac{(j+1)a_{j+1}-ja_j}{\gamma_j}$. Then for any $X \geq j+1$ and $j \leq n-1$ we have:

$$\gamma_j(e_j - X) \leq \gamma_j(e_j - (j+1))$$
$$= \gamma_j\left(2\frac{(j+1)a_{j+1}-ja_j}{\gamma_j} - (j+1)\right)$$
$$= 2(j+1)a_{j+1} - 2ja_j - (j+1)\gamma_j$$
$$= 2(j+1)a_{j+1} - 2ja_j - (j+1)(a_{j+1} - a_j)$$
$$= (j+1)a_{j+1} - 2ja_j + (j+1)a_j$$
$$= \frac{-(d-t_{j+1})}{j+1} + 2\frac{(d+t_{j+1})}{j} - (j+1)\frac{d+t_{j+1}}{j^2}$$
$$= \frac{-j^2(d+t_{j+1}) + 2j(j+1)(d+t_j) - (j+1)^2(d+t_j)}{j^2(j+1)}$$
$$= \frac{-(j^2 - 2j(j+1) + (j+1)^2)d - j^2 t_j + 2j(j+1)t_j}{j^2(j+1)}$$
$$= \frac{-(j-(j+1))^2 d - j^2 t_j + 2j(j+1)t_j}{j^2(j+1)}$$
$$= \frac{-d - j^2 t_j + 2j(j+1)t_j}{j^2(j+1)}$$
$$= \frac{-d}{j^2(j+1)} + \frac{-j^2 t_j + 2j(j+1)t_j}{j^2(j+1)}$$

So we see that as $d$ increases, this bound guarantees that $\gamma_j(e_j - X)$ becomes more negative for each $X \geq j+1$. Also note that for the special zero-neuron, for $X \geq 1$ the net input will be $-MX + t_0 + d \leq -M + t_0 + d$, which will shrink as $M$ grows.

For neurons belonging to building block $j$ which have a positive valued input, we have that $X < e_j$. Note that for any $X \leq j$ and $j < n$ we have:

$$\gamma_j(e_j - X) \geq \gamma_j(e_j - j) = \gamma_j\left(2\frac{(j+1)a_{j+1}-ja_j}{\gamma_j} - j\right)$$
$$= 2(j+1)a_{j+1} - 2ja_j - j\gamma_j$$
$$= 2(j+1)a_{j+1} - 2ja_j - j(a_{j+1} - a_j)$$
$$= 2(j+1)a_{j+1} - ja_j - ja_{j+1}$$
$$= 2\frac{-(d+t_{j+1})}{j+1} + \frac{(d+t_j)}{j} + j\frac{(d+t_{j+1})}{(j+1)^2}$$
$$= \frac{-2j(j+1)(d+t_{j+1}) + (j+1)^2(d+t_j) + j^2(d+t_{j+1})}{j(j+1)^2}$$
$$= \frac{((j+1)^2 - 2j(j+1) + j^2)d + (j+1)^2 t_j - 2j(j+1)t_{j+1} + j^2 t_{j+1}}{j(j+1)^2}$$
$$= \frac{(j+1-j)^2 d + (j+1)^2 t_j - 2j(j+1)t_{j+1} + j^2 t_{j+1}}{j(j+1)^2}$$
$$= \frac{d + (j+1)^2 t_j - 2j(j+1)t_{j+1} + j^2 t_{j+1}}{j(j+1)^2}$$
$$= \frac{d}{j(j+1)^2} + \frac{(j+1)^2 t_j - 2j(j+1)t_{j+1} + j^2 t_{j+1}}{j(j+1)^2}$$

And for the case $j = n$, we have for $X \leq j$ that:

$$\gamma_j(e_j - X) \geq \gamma_j(e_j - j) = \frac{d+t_n}{n^2}(2n - n) = \frac{d}{n} + \frac{t_n}{n}$$

So in all cases we see that as $d$ increases, this bound guarantees that $\gamma_j(e_j - X)$ grows linearly. Also note that for the special zero-neuron, the net input will be $t_0 + d$ for $X = 0$, which will grow linearly as $d$ increases.

$\square$

## A.6 Proofs for Section 4

### A.6.1 Proof of Theorem 8

We first state some basic facts which we need.

**Fact 14** (Muroga (1971)). *Let $f : \{0,1\}^n \to \{0,1\}$ be a Boolean function computed by a threshold neuron with arbitrary real incoming weights and bias. There exists a constant $K$ and another threshold neuron computing $f$, all of whose incoming weights and bias are integers with magnitude at most $2^{Kn \log n}$.*

A direct consequence of the above fact is the following fact, by now folklore, whose simple proof we present for the sake of completeness.

**Fact 15.** *Let $f_n$ be the set of all Boolean functions on $\{0,1\}^n$. For each $0 < \alpha < 1$, let $f_{\alpha,n}$ be the subset of such Boolean functions that are computable by threshold networks with one hidden layer with at most $s$ neurons. Then, there exits a constant $K$ such that,*

$$\left| f_{\alpha,n} \right| \leq 2^{K(n^2 s \log n + s^2 \log s)}.$$

*Proof.* Let $s$ be the number of hidden neurons in our threshold network. By using Fact 14 repeatedly for each of the hidden neurons, we obtain another threshold network having still $s$ hidden units computing the same Boolean function such that the incoming weights and biases of all hidden neurons is bounded by $2^{Kn \log n}$. Finally applying Fact 14 to the output neuron, we convert it to a threshold gate with parameters bounded by $2^{Ks \log s}$. Henceforth, we count only the total number of Boolean functions that can be computed by such threshold networks with integer weights. We do this by establishing a simple upper bound on the total number of distinct such networks. Clearly, there are at most $2^{Kn^2 \log n}$ ways to choose the incoming weights of a given neuron in the hidden layer. There are $s$ incoming weights to choose for the output threshold, each of which is an integer of magnitude at most $2^{Ks \log s}$. Combining these observations, there are at most $2^{Ks \cdot n^2 \log n} \times 2^{Ks^2 \log s}$ distinct networks. Hence, the total number of distinct Boolean functions that can be computed is at most $2^{K(n^2 s \log n + s^2 \log s)}$. $\square$

With these basic facts in hand, we prove below Theorem 8 using Proposition 5 and Theorem 6.

*Proof of Theorem 8.* Consider any thresholded RBM network with $m$ hidden units that is computing a $n$-dimensional Boolean function with margin $\delta$. Using Proposition 5, we can obtain a thresholded hardplus RBM network of size $4m^2/\delta \cdot \log(2m/\delta) + m$ that computes the same Boolean function as the thresholded original RBM network. Applying Theorem 6 and thresholding the output, we obtain a thresholded network with 1 hidden layer of thresholds which is the same size and computes the same Boolean function. This argument shows that the set of Boolean functions computed by thresholded RBM networks of $m$ hidden units and margin $\delta$ is a subset of the Boolean functions computed by 1-hidden-layer threshold networks of size $4m^2 n/\delta \cdot \log(2m/\delta) + mn$. Hence, invoking Fact 15 establishes our theorem.

$\square$

### A.6.2 Proof of Theorem 9

Note that the theorems from Hajnal et al. (1993) assume integer weights, but this hypthosis can be easily removed from their Theorem 3.6. In particular, Theorem 3.6 assumes nothing about the lower weights, and as we will see, the integrality assumption on the top level weights can be easily replaced with a margin condition.

First note that their Lemma 3.3 only uses the integrality of the upper weights to establish that the margin must be $\geq 1$. Otherwise it is easy to see that with a margin $\delta$, Lemma 3.3 implies that a threshold neuron in a thresholded network of size $m$ is a $\frac{2\delta}{\alpha}$-discriminator ($\alpha$ is the sum of the

absolute values of the 2nd-level weights in their notation). Then Theorem 3.6's proof gives $m \geq \delta 2^{(1/3-\epsilon)n}$ for sufficiently large $n$ (instead of just $m \geq 2^{(1/3-\epsilon)n}$). A more precise bound that they implictly prove in Theorem 3.6 is $m \geq \frac{6\delta 2^{n/3}}{C}$.

Thus we have the following fact adapted from Hajnal et al. (1993):

**Fact 16.** *For a neural network of size $m$ with a single hidden layer of threshold neurons and weights bounded by $C$ that computes a function that represents $IP$ with margin $\delta$, we have $m \geq \frac{6\delta 2^{n/3}}{C}$.*

*Proof of Theorem 9.* By Proposition 5 it suffices to show that no thresholded hardplus RBM network of size $\leq 4m^2 \log(2m/\delta)/\delta + m$ with parameters bounded by $C$ can compute $IP$ with margin $\delta/2$.

Well, suppose by contradiction that such a thresholded RBM network exists. Then by Theorem 6 there exists a single hidden layer threshold network of size $\leq 4m^2 n \log(2m/\delta)/\delta + mn$ with weights bounded in magnitude by $(n+1)C$ that computes the same function, i.e. one which represents $IP$ with margin $\delta/2$.

Applying the above Fact we have $4m^2 n \log(2m/\delta)/\delta + mn \geq \frac{3\delta 2^{n/3}}{(n+1)C}$.

It is simple to check that this bound is violated if $m$ is bounded as in the statement of this theorem.

$\square$

### A.6.3 Proof of Theorem 10

We prove a more general result here from which we easily derive Theorem 10 as a special case. To state this general result, we introduce some simple notions. Let $h : \mathbb{R} \to \mathbb{R}$ be an *activation function*. We say $h$ is monotone if it satisfies the following: Either $h(x) \leq h(y)$ for all $x < y$ OR $h(x) \geq h(y)$ for all $x < y$. Let $\ell : \{0,1\}^n \to \mathbb{R}$ be an inner function. An $(h, \ell)$ gate/neuron $G_{h,\ell}$ is just one that is obtained by composing $h$ and $\ell$ in the natural way, i.e. $G_{h,\ell}(x) = h(\ell(x))$. We notate $\left|\left|(h,\ell)\right|\right|_\infty = \max_{x \in \{0,1\}^n} \left|G_{h,\ell}(x)\right|$.

We assume for the discussion here that the number of input variables (or observables) is even and is divided into two halves, called $x$ and $y$, each being a Boolean string of $n$ bits. In this language, the inner production Boolean function, denoted by $IP(x,y)$, is just defined as $x_1 y_1 + \cdots + x_n y_n$ (mod 2). We call an inner function of a neuron/gate to be $(x,y)$-separable if it can be expressed as $g(x) + f(y)$. For instance, all affine inner functions are $(x,y)$-separable. Finally, given a set of activation functions $H$ and a set of inner functions $I$, an $(H,I)$- network is one each of whose hidden unit is a neuron of the form $G_{h,\ell}$ for some $h \in H$ and $\ell \in I$. Let $\left|\left|(H,I)\right|\right|_\infty = \sup\left\{\left|\left|(h,\ell)\right|\right|_\infty : h \in H, \ell \in I\right\}$.

**Theorem 17.** *Let $H$ be any set of monotone activation functions and $I$ be a set of $(x,y)$ separable inner functions. Then, every $(H,I)$ network with one layer of $m$ hidden units computing IP with a margin of $\delta$ must satisfy the following:*

$$m \geq \frac{\delta}{2\left|\left|(H,I)\right|\right|_\infty} 2^{n/4}.$$

In order to prove Theorem 17, it would be convenient to consider the following 1/-1 valued function: $(-1)^{\text{IP}(x,y)} = (-1)^{x_1 y_1 + \cdots + x_n y_n}$. Please note that when IP evaluates to 0, $(-1)^{\text{IP}}$ evaluates to 1 and when IP evaluates to 1, $(-1)^{\text{IP}}$ evaluates to -1.

We also consider a matrix $M_n$ with entries in $\{1, -1\}$ which has $2^n$ rows and $2^n$ columns. Each row of $M_n$ is indexed by a unique Boolean string in $\{0,1\}^n$. The columns of the matrix are also indexed similarly. The entry $M_n[x,y]$ is just the 1/-1 value of $(-1)^{\text{IP}(x,y)}$. We need the following fact that is a special case of the classical result of Lindsey.

**Lemma 18** (Chor and Goldreich,1988)**.** *The magnitude of the sum of elements in every $r \times s$ submatrix of $M_n$ is at most $\sqrt{rs 2^n}$.*

We use Lemma 18 to prove the following key fact about monotone activation functions:

**Lemma 19.** *Let $G_{h,\ell}$ be any neuron with a monotone activation function $h$ and inner function $\ell$ that is $(x,y)$-separable. Then,*

$$\left|\mathbb{E}_{x,y}\left[G_{h,\ell}\big(x,y\big)(-1)^{IP\big(x,y\big)}\right]\right| \leq ||(h,\ell)||_\infty \cdot 2^{-\Omega(n)}. \tag{2}$$

*Proof.* Let $\ell(x,y) = g(x) + f(y)$ and let $0 < \alpha < 1$ be some constant specified later. Define a total order $\prec_g$ on $\{0,1\}^n$ by setting $x \prec_g x'$ whenever $g(x) \leq g(x')$ and $x$ occurs before $x'$ in the lexicographic ordering. We divide $\{0,1\}^n$ into $t = 2^{(1-\alpha)n}$ groups of equal size as follows: the first group contains the first $2^{\alpha n}$ elements in the order specified by $\prec_g$, the second group has the next $2^{\alpha n}$ elements and so on. The $i$th such group is denoted by $X_i$ for $i \leq 2^{(1-\alpha)n}$. Likewise, we define the total order $\prec_f$ and use it to define equal sized blocks $Y_1, \ldots, Y_{2^{(1-\alpha)n}}$.

The way we estimate the LHS of (2) is to pair points in the block $(X_i, Y_j)$ with $(X_{i+1}, Y_{j+1})$ in the following manner: wlog assume that the activation function $h$ in non-decreasing. Then, $G_{h,\ell}(x,y) \leq G_{h,\ell}(x',y')$ for each $(x,y) \in (X_i, Y_j)$ and $(x',y') \in (X_{i+1}, Y_{j+1})$. Further, applying Lemma 18, we will argue that the total number of points in $(X_i, Y_j)$ at which the product in the LHS evaluates negative (positive) is very close to the number of points in $(X_{i+1}, Y_{j+1})$ at which the product evaluates to positive (negative). Moreover, by assumption, the composed function $(h, \ell)$ does not take very large values in our domain by assumption. These observations will be used to show that the points in blocks that are diagonally across each other will almost cancel each other's contribution to the LHS. There are too few uncancelled blocks and hence the sum in the LHS will be small. Forthwith the details.

Let $P_{i,j}^+ = \{(x,y) \in (X_i, Y_j) \,|\, IP(x,y) = 1\}$ and $P_{i,j}^- = \{(x,y) \in (X_i, Y_i) \,|\, IP(x,y) = -1\}$. Let $t = 2^{(1-\alpha)n}$. Let $h_{i,j}$ be the max value that the gate takes on points in $(X_i, Y_j)$. Note that the non-decreasing assumption on $h$ implies that $h_{i,j} \leq h_{i+1,j+1}$. Using this observation, we get the following:

$$\mathbb{E}_{x,y}\left[G_{h,\ell}\big(x,y\big)(-1)^{IP\big(x,y\big)}\right] \leq \frac{1}{4^n}\left|\sum_{(i,j)<t} h_{i,j}\left(\big|P_{i,j}^+\big| - \big|P_{i+1,j+1}^-\big|\right)\right| + \frac{1}{4^n}\sum_{i=t\,\mathrm{OR}\,j=t} h_{i,j}|P_{i,j}| \tag{3}$$

We apply Lemma 18 to conclude that $\big|P_{i+1,j+1}^+\big| - \big|P_{i,j}^-\big|$ is at most $2 \cdot 2^{(\alpha+1/2)n}$. Thus, we get

$$\text{RHS of (3)} \leq ||(h,\ell)||_\infty \cdot \left(2 \cdot 2^{-(\alpha-\frac{1}{2})n} + 4 \cdot 2^{-(1-\alpha)n}\right). \tag{4}$$

Thus, setting $\alpha = 3/4$ gives us the bound that the RHS above is arbitrarily close to $||(h,\ell)||_\infty \cdot 2^{-n/4}$.

Similarly, pairing things slightly differently, we get

$$\mathbb{E}_{x,y}\left[G_{h,\ell}\big(x,y\big)(-1)^{IP\big(x,y\big)}\right] \geq \frac{1}{4^n}\sum_{(i,j)<t} h_{i+1,j+1}\left(\big|P_{i+1,j+1}^+\big| - \big|P_{i,j}^-\big|\right) - \frac{1}{4^n}\sum_{i=t\,\mathrm{OR}\,j=t} |h_{i,j}| \cdot |P_{i,j}| \tag{5}$$

Again similar conditions and settings of $\alpha$ imply that RHS of (5) is no smaller than $-||(h,\ell)||_\infty \cdot 2^{-n/4}$, thus proving our lemma. $\qquad\square$

We are now ready to prove Theorem 17.

**Proof** *of Theorem 17.* Let $C$ be any $(H,I)$ network having $m$ hidden units, $G_{h_1,\ell_1}, \ldots, G_{h_m,\ell_m}$, where each $h_i \in H$ and each $\ell_i \in I$ is $(x,y)$-separable. Further, let the output threshold gate be such that whenever the sum is at least $b$, $C$ outputs 1 and whenever it is at most $a$, $C$ outputs -1. Then, let $f$ be the sum total of the function feeding into the top threshold gate of $C$. Define $t = f - (a+b)/2$. Hence,

$$\mathbb{E}_{x,y}\big[f(x,y)(-1)^{IP(x,y)}\big] = \mathbb{E}_{x,y}\big[t(x,y)(-1)^{IP}(x,y)\big] + \frac{a+b}{2}\mathbb{E}_{x,y}\big[(-1)^{IP(x,y)}\big]$$

$$\geq (b-a)/2 + \frac{a+b}{2}\mathbb{E}_{x,y}\big[(-1)^{IP(x,y)}\big].$$

Thus, it follows easily

$$\left| \mathbb{E}_{x,y}\left[ f(x,y)(-1)^{\mathrm{IP}(x,y)} \right] \right| \geq \frac{b-a}{2} - \frac{|a+b|}{2} 2^{-n}. \tag{6}$$

On the other hand, by linearity of expectation and applying Lemma 19, we get

$$\left| \mathbb{E}_{x,y}\left[ f(x,y)(-1)^{\mathrm{IP}(x,y)} \right] \right| \leq \sum_{j=1}^{m} \left| \mathbb{E}_{x,y}\left[ G_{h_j,\ell_j}(x,y)(-1)^{\mathrm{IP}(x,y)} \right] \right| \leq m \cdot \left\| (H,I) \right\|_{\infty} \cdot 2^{-n/4}. \tag{7}$$

Comparing (6) and (7), observing that each of $|a|$ and $|b|$ is at most $m\left\|(H,I)\right\|_{\infty}$ and recalling that $\delta = (b-a)$, our desired bound on $m$ follows. $\qquad \square$

*Proof of Theorem 10.* The proof follows quite simply by noting that the set of activation functions in this case is just the singleton set having only the monotone function $soft(y) = \log(1+\exp(y))$. The set of inner functions are all affine functions with each coefficient having value at most $C$. As the affine functions are $(x,y)$-separable, we can apply Theorem 17. We do so by noting $\left\|(H,I)\right\|_{\infty} \leq \log(1+\exp(nC)) \leq \max\{\log 2, nC + \log 2\}$. That yields our result. $\qquad \square$

**Remark 20.** *It is also interesting to note that Theorem 17 appears to be tight in the sense that none of the hypotheses can be removed. That is, for neurons with general non-montonic activation functions, or for neurons with monotonic activation functions whose output magnitude violates the aforementioned bounds, there are example networks that can efficiently compute any real-valued function. Thus, to improve this result (e.g. removing the weight bounds) it appears one would need to use a stronger property of the particular activation function than monotonicity.*

