[Reviews · NeurIPS 2013]

Submitted by Assigned_Reviewer_5

This paper shows good progress in understanding Restricted Boltzmann Machines from the perspective expressive power of neural network. One interesting result is that RBM can efficiently implement parity function, but not more complicated functions, assuming an exponential bound on the weights.

One fact that may weaken the contribution of this paper is that previous works have suggested that the number of hidden units is exponential to the cost. For example, in Montufar et al's NIPS11, it is proved that the KL divergence is bounded by (n-1) - log (m-1). However, the new discovery in this submission provides more insights. As suggested by the authors, in the future it will be truly exciting to explore the relationship between the number of hidden unit and the number of layers in deep RBMs.

This is a theoretical paper and there is no experiment.
Summary: This paper has the potentials of contributing to a breakthrough in understanding Restricted Boltzmann Machines.

Submitted by Assigned_Reviewer_6

The paper provides a theoretical analysis of the types of distributions that can efficiently be represented by restricted Boltzmann machines (RBMs). The analysis is based on a representation of the unnormalized log probability (free energy) of RBMs as a special form of neural network (NN). The paper relates these RBM networks to more common types of NNs whose properties have been studied in the literature. This approach allows the authors to identify two non-trivial examples of functions that can and cannot be represented efficiently by RBM networks - and hence related distributions can / cannot be modeled efficiently by RBMs. Specifically they show that RBM networks can efficiently represent any function that only depends on the number of non-zero visible units, such as parity, but that they are unable to represent the only somewhat more difficult example of inner product parity.


I think the paper addresses an interesting and important problem: RBMs have received a lot of attention in the recent literature but their properties are still not well understood. Many applications of RBMs refer to the result that RBMs are universal approximators of binary distributions, but these results give no indication what distributions can be *efficiently* represented. This question is largely unanswered.


Against this background, the paper makes two interesting contributions:

Firstly, although the paper clearly does not "solve" the problem of characterizing the efficiently representable distributions I think it nevertheless adds a useful piece to the puzzle: It provides two concrete examples of meaningful and non-trivial distributions that can and cannot be represented efficiently by RBMs. These results are, in my eyes, different and complementary to previous works on this topic, which have either focused on the universal approximator properties of RBMs, or have characterized the set of distributions that can be represented by RBMs in a very abstract manner.

Secondly, the method of the proof, which is based on an analysis of the free energy function as a special type of NN, is quite different from the approaches taken in previous works that I have seen: By establishing connections between the special type of RBM networks and NN more widely used in the literature the paper makes it possible to apply established results from the analysis of such networks. This (to my knowledge) new perspective is interesting and might enable further progress in understanding RBMs and related architectures in the future.

Despite its theoretical nature I find the paper to be well written. It provides a good overview of previous work on the representational capabilities of RBMs. The results of the paper are well structured and explained, and should be accessible to most of the NIPS community. One concern is, however, that all proofs had to be deferred to the appendix, so maybe it'd be more appropriate to turn this into a longer journal paper -- where the technical aspects can be properly checked (I only skimmed some of the supplemental material).


Other points:
> Maybe, a few more words could be spent on explaining why the particular functions being analyzed in sections 3 and 4 are interesting examples to study. For IP you have a sentence dedicated to this question (lines 382-384), but I think it would be useful to expand on this.

> I believe the universality results, for which you cite Le Roux & Bengio (2008) have already been established earlier in the tech report by Freund & Haussler (1994): "Unsupervised learning of distributions of binary vectors using two layer networks" (UCSC-CRL-94-25)

> I am wondering to what extent it is going to be possible to adapt the analysis technique in the paper to deeper architectures (e.g. Deep Boltzmann Machines). Here, the latent variable cannot be marginalized analytically so it is not immediately obvious to me how they could be translated into easy-to-analyze networks.

Summary: The authors address an interesting and relevant question: which distributions can be efficiently represented by RBMs? It is quite dense but still well written and provides a new view on the problem, with results complementary to previous works on this problem.

Submitted by Assigned_Reviewer_7

The authors proposed two modified RBM models, i.e. softplus RBM and hardplus RBM, and proved that for every symmetric function f defined on {0, 1}^n there exists a hardplus RBM with n^2+1 hidden units that can computes f exactly. They also proved that the "inner product mod 2" function can not be computed by a hardplus or softplus RBM with less than exponential number of hidden units.

Quality: There are many useful theorems in this paper which bound the number of hidden units and approximation error. The authors first prove that hardplus RBM can compute a symmetric function well and analysis the approximation error of the softplus RBM through the differences between hardplus and softplus RBM. It's a complete piece of work in theory.

Clarity: The paper is well-orgnazed and clearly written except for some spell errors.

Originality: These two models are not proposed by the authors first, yet, to our knowledge it's the first analytical result about the expression power of them on symmetric functions.

Significance: The authors were trying to compute a symmetric function which f(x1,...,xn)=g(x1+...+xn). We can convert x1+...+xn into a binary expression as y1y2...yp, where y1, y2, ..., yp are 0-1 variables. So p is approximately O(log(n)), and if we want to compute a function on {0,1}^p, we need at most O(2^p)(approximately O(n)) hidden units to compute any functions on y1 through yp. We can see that symmetry constraint is very strong so we don't think this work is very significant.
Summary: The symmetry constraint is too strong and fade the work even though there many useful theorems in the paper.
Author Feedback

Author rebuttal: We thank the reviewers for their comments and suggestions. Below we give an in-depth response to each review.

Reviewer 5:

We would like to clarify the relationship between our work and the previous work on RBMs and particularly Montufar et al. While this paper considers the same topic as that work, it is best viewed not as an extension of it, but as an application of the results and techniques first developed by circuit theorists and in particular the Hajnal et al. paper.

Montufar et al. show that RBMs can implement what are essentially disjoint mixtures of product distributions. Basically, an input x is assigned a probability based on which one of m groups it falls into (m = number of hiddens) according to whether certain entries of x match to certain “templates”. Once the group is determined, the likelihood factorizes over the inputs (i.e., they are treated as having no dependency). While this is an interesting result, it cannot be used to show that RBMs can implement multi-layer circuit-style “computations” for their log-likelihoods, as our symmetric function construction does.

The KL divergence result in Montufar et al is not a lower bound, and provides only a loose upper bound which says the error is bounded from above, by essentially having the RBM memorize its training data (without exploiting any structure that exists in the distribution). So it shows that exponentially many units are sufficient to model arbitrary distributions, but says nothing about how many may be necessary.

To the best of our knowledge, there are no prior results that can bound any reasonable measure of best-achievable approximation error *from below* by a function of m, thus showing that a certain size of m is necessary to achieve 0 error. Without our IP hardness result, it was actually conceivable that for any poly-time computable function, the best achievable KL with an RBM could be 0 for m = n (versus the m=2^(n-1)). By lower bounding m for a very easy to compute function like IP, we formally rule this out, assuming a realistic bound on the size of the weights. This result also says something directly about the hardness of functions which IP reduces to (see below).


Reviewer 6:

We agree with your points and will try to improve the paper in the ways you have suggested.

We will expand on the background about the IP function and why it is theoretically interesting, e.g., “There are many computational problems that IP can be reduced to, so showing that RBM networks cannot compute IP thus proves that RBMs cannot efficiently model a wide range of distributions whose unnormalized log-likelihoods are sufficiently “complex” in a computational sense. Examples include distributions whose density involves the computation of functions such as binary multiplication, graph connectivity, and more. Prior to the hardness result in this paper, for the class of efficiently computable (unnormalized) log-likelihood functions there was no formal separation between RBMs and arbitrarily deep circuits, or even between RBMs and an iterative poly-time algorithm.”

We are already working on a paper that examines the expressive power in deeper probabilistic models, and indeed they can compute functions such as IP and more, as one would expect. However, without the results in this paper we wouldn't know for sure if they are actually more powerful than shallow RBMs.


Reviewer 7:

We would like to address your comments about symmetric functions. That symmetric functions can be efficiently computed by RBMs is a surprising and non-trivial result, and was believed to be false by people who have studied expressive power of shallow models (e.g., Yoshua Bengio). This is because symmetric functions like PARITY are highly non-smooth, and straightforward methods for deriving an efficient construction fail.

Computing symmetric functions is a 2-stage process, that involves first the computation of the sum x1+x2+...+xn, followed by the application of an arbitrary function on the output of this sum. The summation step is a critical component of the task of computing symmetric functions, and while it is a computationally simple operation, RBMs are themselves computationally primitive “shallow” models, and so there is no reason a-priori why they should be able to implicitly perform this operation, especially in sequence with another non-trivial operation. Logical Boolean circuits, for example, require non-constant depth to compute the symmetric function PARITY.

A key point is that you cannot feed the answer of this addition operation to the model directly, as you are then no longer modeling a distribution on the same space x, nor are you computing a function of the input as given. performing this kind of preprocessing essentially solves half of the modeling problem. If one is allowed to include arbitrary problem-specific transformations, then virtually any model that has a weak universality property can capture not only symmetric functions but any distribution. For general-purpose models like RBMs we don't know what this hypothetical transformation is ahead of time when applying our models to actual problems,
and thus we are interested in understanding how well these models can do on the original problems *as given*.

Beyond what the result says itself, this is also the first demonstration ever given that RBMs can perform any non-trivial multi-stage computations beyond the table look-up of the Montufar et al. construction.

Finally, it seems unfair to judge our work solely based on the symmetric function construction alone. Its importance is best understood in context of our other results, and in particular the IP hardness result, which shows that if we add just one additional operation to the required computation (i.e., the pair-wise products of the input), the problem can no longer be efficiently solved by RBMs.